# Farm Animal Welfare during Transport and at the Slaughterhouse: Perceptions of Slaughterhouse Employees, Livestock Drivers, and Veterinarians

**DOI:** 10.3390/ani14030443

**Published:** 2024-01-29

**Authors:** Maja Lipovšek, Andrej Kirbiš, Iztok Tomažič, Alenka Dovč, Manja Križman

**Affiliations:** 1Institute for Food Safety, Feed and Environment, Veterinary Faculty, University of Ljubljana, 1000 Ljubljana, Slovenia; majalip11@gmail.com (M.L.); andrej.kirbis@vf.uni-lj.si (A.K.); 2Department of Biology, Biotechnical Faculty, University of Ljubljana, 1000 Ljubljana, Slovenia; iztok.tomazic@bf.uni-lj.si; 3Clinic for Birds, Small Mammals and Reptiles, Veterinary Faculty, University of Ljubljana, 1000 Ljubljana, Slovenia; alenka.dovc@vf.uni-lj.si

**Keywords:** welfare, transport, livestock, slaughterhouse, legislation

## Abstract

**Simple Summary:**

This study investigated the level of knowledge and the current situation with regard to the welfare of farm animals during transportation and in beef, pork, and poultry slaughterhouses. For this purpose, a questionnaire was developed to obtain data on respondents’ understanding of their work, knowledge of legislation, training, and attitudes towards animal welfare. Slaughterhouse employees and professional animal livestock drivers participated in the study. Slaughterhouse employees showed more knowledge about animal welfare than livestock drivers, but both groups were not sufficiently familiar with animal welfare legislation and regulations. All respondents agreed that animals are sentient beings and almost all respondents were unfamiliar with the concept of biosecurity. This study found that the results of the veterinary experts’ observations were generally lower than the results of the employees’ and livestock drivers’ self-assessments. Based on the research findings, it can be concluded that there is a need to improve the awareness and knowledge of slaughterhouse employees and livestock drivers regarding animal handling and welfare regulations. This would include providing better hands-on training, better knowledge of legislation, and raising awareness of the benefits of certain procedures and standards in slaughterhouses and during transportation.

**Abstract:**

Animal welfare is a multidimensional concept that includes several physical and psychological parameters of the animal. The aim of this study was to assess animal welfare during transportation and in Slovenian beef, pork, and poultry slaughterhouses. A questionnaire was used for this study. Several parameters of animal welfare were rated on a 5-point scale, such as health status, animal behavior, lairage or transport vehicle conditions, and driver regulation compliance. The scale was also used for the second part of the study. This consisted of two studies: (1) self-assessment by slaughterhouse employees and livestock transport drivers and (2) animal welfare observational assessment performed by two veterinarians. The results were compared with each other. Ten large slaughterhouses and nine livestock drivers took part in the survey. The results showed that slaughterhouse employees knew more about animal welfare than livestock truck drivers, but both groups were not sufficiently familiar with animal welfare laws and regulations. This study found that the experts’ assessments were generally lower than the self-assessments of employees and livestock drivers. Based on the research findings, it can be concluded that there is a need to improve the awareness and knowledge of slaughterhouse employees and livestock drivers regarding animal handling and animal welfare regulations.

## 1. Introduction

Animal welfare means how an animal copes with the conditions in which it lives. An animal is in a good state of welfare if it is (scientifically proven) healthy, comfortable, well fed, safe, able to act out its innate behavior, and does not suffer from unpleasant conditions such as pain, fear, and distress [1]. Animal welfare requires disease prevention and veterinary treatment, appropriate housing, management, nutrition, humane treatment, and humane slaughter [2,3]. Various concepts of animal welfare have been developed over the last half decade [4].

Farm animals such as poultry, pigs, cows, sheep, goats, and horses are usually under stress due to high production and economic demands. In recent decades, we have gained a better overview of the parameters that influence animal welfare [5].

Protecting animal welfare during the slaughter process is about minimizing the pain, distress, and suffering of farm animals at the time of slaughter. Slaughterhouse staff must therefore apply various procedures. For example, they must regularly check that the animals show no signs of consciousness or sentience between the end of the stunning process and death [6]. Current European Union legislation on the welfare of animals at the time of killing requires that personnel working with live animals have a certificate of competence. Food business operators must ensure that certain slaughter operations, e.g., movement, stunning, and slaughter, are only carried out by persons holding such a certificate. The equipment used for slaughter and the design and construction of the slaughterhouse must also comply with the regulations [7]. In order to obtain the certificate of competence, personnel must complete a training course consisting of practical training on animal handling and correct slaughtering, basic training, and refresher training [8].

Recently, more data have been collected in slaughterhouses to monitor the welfare of animals on the farm, during transportation, and on the slaughterhouse premises [9]. The slaughterhouse can be a dangerous and psychologically stressful workplace [10]. Slaughterhouse employees and livestock drivers must comply with legal regulations and work quickly at the same time. This can be difficult for an employee if they do not have practical knowledge and skills. However, if the slaughterhouse does not have the appropriate facilities or technical equipment, it is even more difficult for slaughterhouse employees to move animals quickly and in accordance with best animal welfare practices.

The objective of our research project was to assess the knowledge of slaughterhouse employees and livestock drivers about animal welfare indicators. In Slovenia, there have been several animal welfare scandals in slaughterhouses in recent years. The aim of our study was therefore to find out whether the problem lies in the knowledge of animal welfare among people working with live animals and what knowledge can be improved. For this purpose, we used a questionnaire with (I) basic information, (II) general statements, and (III) statements about practical work with live animals. In addition, we assessed (IV) the perception of the participants’ practical work on animal welfare based on an actual animal welfare situation in a slaughterhouse lairage.

## 2. Materials and Methods

Ten slaughterhouse employees and nine livestock truck drivers participated in this study (Table 1). Slovenia is divided into ten regions for official control in slaughterhouses (R1–R10 in Table 1) by the Slovenian Administration for Food Safety, Veterinary Sector and Plant Protection. For our study, we selected the largest slaughterhouse in Slovenia and one livestock driver from every selected slaughterhouse who transported animals for slaughter on that day. There are seven slaughterhouses for poultry meat in Slovenia, three small and four large ones. All four large slaughterhouses participated in our study. For red meat, there are 83 slaughterhouses in Slovenia, of which 58 (70%) are small (slaughtering less than 1000 livestock units per year) and 25 (30%) are large. We surveyed six large red meat slaughterhouses, which means that 24% of our sample consists of large slaughterhouses. However, these are the largest slaughterhouses in Slovenia, with the fastest slaughter lines and most cattle and pigs are slaughtered here. In Slovenian slaughterhouses, work is performed only during morning shifts. This means that the slaughterhouse staff usually consisted of two people who were suitable for the interview.

We first contacted the biggest slaughterhouses in Slovenia according to data from the Slovenian Administration for Food Safety, Veterinary Sector and Plant Protection. We contacted them by telephone to find out whether they wanted to participate in the animal welfare research project. If the slaughterhouse agreed, we arranged a date for the visit of veterinary experts. The slaughterhouse manager decided which slaughterhouse employee would take part in the project. However, the selected employee was asked by the interviewing veterinarian if they were willing to participate and, if so, they were presented with a consent form to sign. All slaughterhouse employees who participated in the study had a certificate of competence in accordance with Regulation (EC) 1099/2009. To obtain the certificate of competence, the employee must complete a theoretical and practical part of the training. The first part is completed in a company accredited by the Slovenian Administration for Food Safety, Veterinary Sector and Plant Protection. The theoretical part covers the basics of animal welfare, animal welfare legislation, species-specific characteristics of animal movement, and methods of stunning and killing. After the theoretical part, employees must pass a written examination. The second, practical part takes place in the slaughterhouse where the employees work [11]. After the interview with the slaughterhouse employee, we asked the first livestock truck driver who brought the animals to the slaughterhouse that day whether he would like to participate in the research project. If he agreed, a consent form was presented to him for signature. In addition, all livestock truck drivers participating in the study were professional drivers for live animals and were trained in accordance with Regulation (EC) 1/2005. In one slaughterhouse R2, only local farmers were transporting livestock to the slaughterhouse on the day of the survey. As they do not require certification under Regulation (EC) 1/2005, they did not meet the criteria for the project.

The sample size of the red meat slaughterhouses is a limitation. However, 10 large slaughterhouses were contacted, 4 of which declined to participate for various reasons. The speed of the slaughter lines in the 15 remaining slaughterhouses is much slower than in the selected ones, which is why they were not included in the study.

### 2.1. Questionnaire with Protocol

For this study, an animal welfare questionnaire with protocol was created for slaughterhouse employees and livestock drivers (AWQ/SL and AWQ/D) (Appendix A). General data and participant demographics were collected first, followed by 29 items assessing participants’ general beliefs about animal welfare: (1) the importance of information (seven items); (2) the influence of the public on animal welfare (five items); (3) animal welfare (five items); (4) the utilitarian/dominionistic view of farm animals (five items); and (5) the relationship between humans and animals (seven items). The third part of the questionnaire consisted of 25 questions about the participants’ knowledge of animal welfare: (1) what animal welfare is, (2) water and feed requirements, (3) animal well being, and (4) health status. The protocol consisted of questions on the perceived importance of the participants and related observational assessment questions: (1) general status, (2) animal behavior, (3) lairage conditions, and (4) environmental conditions. The questionnaires on the importance of animal welfare at the slaughterhouse and during transport (self-assessment) were identical and were used to compare the assessment of the importance of animal welfare at slaughter and during transport. Observational assessment parameters were rated on a 5-point scale, while employee/driver perceived importance was rated on a 5-point Likert scale.

Two veterinarians familiar with slaughter and transportation assessed the welfare protocol. Both were previously trained by the project leader, a diplomate of the European College of Animal Welfare and Behavioral Medicine. The observation scoring points have the following meaning: (1) major deficiencies (immediate action required); (2) deficiencies warranting a warning; (3) minor deficiencies (advice required); (4) no deficiencies (compliance with standards); and (5) no deficiencies (above standards). Additional descriptions were provided for each observation point (Appendix A). The legal basis for establishing the point scale was Council Regulation (EC) No. 1/2005, Council Regulation (EC) No. 1099/2009, and the Animal Protection Act from the Official Gazette of the Republic of Slovenia, No. 38/13.

For example, the questions in the questionnaire designed to measure the importance of animal welfare for slaughterhouse employees/livestock drivers began with the question “In your opinion, how important is it that you move the animals quickly but in accordance with the law?” The scale indicates the degree of importance for slaughterhouse employees/livestock drivers: (1) not at all important, (2) not important, (3) undecided, (4) important, and (5) very important.

First, animal welfare was assessed using the protocol, and then slaughterhouse employees/livestock driver was asked about their views on welfare using a questionnaire. Slaughterhouses and livestock drivers were interviewed from 4 April to 14 September 2022.

### 2.2. Statistical Analysis

All raw data were first transferred to Microsoft Excel version 2312 build 16.0.17126.20132 and transformed for use in Statistical Package for the Social Sciences—SPSS (ver. 26). Mean values were calculated for each parameter of the questionnaire (general status, animal behavior, lairage/transport conditions, and environmental conditions) and these were compared between slaughterhouse employees and livestock drivers (Mann–Whitney test). In addition, the Wilcoxon test was applied to compare the observation results with the slaughterhouse employees/livestock drivers’ perceived importance of animal welfare from the protocol and the questionnaire. Due to the small sample size, effect sizes were calculated to determine the strength of statistical differences using the formula r = Z/√N. Values < −0.2 or >0.2 were treated as significant [12].

## 3. Results

Ten slaughterhouse employees and nine livestock drivers took part in this study. The interview lasted an average of 55 to 75 min per participant, including observation by two experts of how the participants carry out their daily work in the field of animal welfare. The gender, age and education level of the participants are summarized in Table 2.

### 3.1. Animal Welfare Beliefs and Awareness Comparison between Slaughterhouse Employees and Livestock Drivers

We compared slaughterhouse employees’ and livestock drivers’ perceptions of animal welfare (Figure 1) and found that slaughterhouse employees generally expressed stronger beliefs than livestock drivers. Although all calculated *p*-values were above the statistical significance level (all *p* > 0.05), the calculated effect sizes indicated that there were no differences in the table for 12 of 29 items (see Appendix A). For four out of seven items assessing the importance of information (Figure 1, A section), slaughterhouse employees rated three statements more positively than livestock drivers. On the other hand, livestock drivers were more likely to think that animal welfare should be taught in schools than slaughterhouse employees (A_06). Several differences were also found in utilitarian/dominionistic attitudes towards farm animals. For example, three out of five items were rated by slaughterhouse employees as more in favor of animal welfare (Figure 1, D section). These participants do not see animals only as a source of income (D_01), do not believe that animals should be treated like machines (D_02), and do not think that sick animals should be killed by the farmers themselves (D_05). However, livestock drivers agree with the statement that sick animals should be euthanized by farmers. In the “animal welfare” category (Figure 1, C-section), two of the five items were rated differently by slaughterhouse employees and livestock drivers. Slaughterhouse employees were more likely to agree that animals should be housed appropriately (C_04) and that farm animals do not have a lower pain threshold (C_05). In the two remaining categories, public influence on animal welfare (Figure 1, B-section) and human–animal relations (Figure 1, E section), only one item each was rated differently by slaughterhouse employees and livestock drivers. Slaughterhouse employees were more likely to agree with the statement that the government should provide financial support to improve animal welfare in slaughterhouses (B_04) and were less likely to agree with the statement that animals only obey when they are afraid of their owners (E_04). Livestock drivers, on the other hand, were rather undecided on this parameter.

Next, we compared the knowledge of animal welfare between slaughterhouse employees and livestock drivers on four different topics (see Appendix A): what is animal welfare (A), water and feed requirements (B), animal welfare (C), and health status (D) (Figure 2). For three statements, the calculated *p*-values were statistically significant (*p* < 0.05). With regard to animal welfare (Figure 2, A section), slaughterhouse employees were undecided on the statement “due to domestication, farm animals have different needs than their ancestors living in the wild”, while livestock drivers agreed with the statement (A_05). The statement that the welfare of the animal is taken care of when it is growing and in good physical condition was also rated higher by the livestock drivers (A_04). On the other hand, slaughterhouse employees rated the statement that we comply with animal welfare standards if the animal reproduces successfully higher (A_03). In section (B), water and feed requirements (Figure 2, B-section), all calculated values were above statistical significance. However, slaughterhouse employees were undecided on the statement (B_04) ‘‘farm animals do not need bedding in the slaughter pens”. Livestock drivers were more likely to answer, “disagree with the statement”. In the section on animal welfare (Figure 2, C section), there was a statistically significant difference in the statement “farm animals are sentient living beings”, which slaughterhouse employees agreed with more strongly than livestock drivers (C_01). Slaughterhouse employees also rated the statements about mobile slaughterhouses and the advantages of slaughtering in regular slaughterhouses positively (C_05). Slaughterhouse employees tended to agree with both methods, while livestock drivers were undecided. In the last section (Figure 2, D-section) on health status, there was a statistically significant difference in the statement “the presence of disease does not affect animal welfare”. Slaughterhouse employees disagreed with this statement, while livestock drivers were undecided (D_01). The most interesting finding was (D_06) that most slaughterhouse employees and livestock drivers were undecided about the statement “The term biosecurity means that the feed for the animals is sanitary”. For the statement “The presence of diseases does not affect the meat products”, both parties disagreed, but slaughterhouse employees agreed more than livestock drivers (D_07).

### 3.2. Comparison of Perceived and Actual Animal Welfare Standards in Slaughterhouse and on the Transport

#### 3.2.1. Observational Assessment by the Veterinary Expert Compared to Self-Assessed Importance by the Slaughterhouse Employees

In the observational assessment by the veterinary experts, the highest average score for the slaughterhouse was awarded for the lairage conditions parameter and the lowest for the environmental conditions (Figure 3). The highest values for the self-assessed importance of animal welfare by the slaughterhouse employees were awarded for the two questions “In your opinion, how important is…”, “…the number of animals per unit area”, and “…that ventilation and heating are regulated/arranged in the lairage?” The comparison of the results of the veterinary expert’s observational assessment with the slaughterhouse employees’ self-assessment of the importance of animal welfare revealed statistically significant differences for five out of ten items (Wilcoxon test, all *p* < 0.05). Moreover, for all but one item, the calculated effect sizes were substantial and ranged from 0.42 to 0.84. In all these cases, the self-assessed importance of animal welfare was rated significantly higher than the score obtained from the veterinary expert’s observational assessment. The most significant difference was clearest in the parameters for handling animals in the lairage (B3). This was followed by the parameters of stocking density in the pens (C1), the way employees enter the lairage (A1), the behavioral condition of the animals (B1 and B2), and the presence of thermometers and hygrometers in the lairage (D5).

#### 3.2.2. Observational Assessment by the Veterinary Expert Compared to Self-Assessed Importance by the Livestock Drivers

In the observational assessment by the veterinary experts of transport, the highest average result was for the “compliance with regulations” factor and the lowest for the general condition and environment (Figure 4). The highest values for the self-assessed importance of animal welfare by the livestock drivers were obtained for the two questions “In your opinion, how important is it…”, “…that you put the correct departure date and time on your driver statement”, and “…that you have a driving license in accordance with Regulation (EC) 1/2005?”. The comparison of the results of the observational assessment with the self-assessed importance of animal welfare revealed statistically significant differences for four out of thirteen items (Wilcoxon test, all *p* < 0.05). For nine of thirteen items, the calculated effect sizes were also considerable and ranged from 0.42 to 0.82. With the exception of two items, the self-assessed importance of animal welfare was significantly higher in all these cases than the score obtained from the observational assessment. However, for the parameter “compliance with regulations” in response to the question “How important do you think it is that you have not been fined for a violation of Regulation (EC) 1/2005?”, the observation score was higher than the self-assessed importance (E6). This is due to the fact that one driver stated that it was not important whether he had received a penalty for the inappropriate transportation of live animals to the slaughterhouse. This is because he received a fine for not complying with the regulations for the transportation of live animals. However, apart from the fine, there were no consequences for him such as, for example, the ban on transporting live animals.

## 4. Discussion

Animal welfare is clearly a concept that can be studied scientifically, but our understanding of animal welfare, and even the science we do to assess and improve animal welfare, is influenced by value-based ideas about what is important or desirable for animals to live well [1].

Our comparison of the opinions of slaughterhouse employees and livestock drivers on animal welfare shows that there are significant differences in their perceptions. Slaughterhouse employees generally have a stronger opinion, while livestock drivers are more indecisive in their responses. A key finding was the statement about the need for practical training to ensure animal welfare. Slaughterhouse employees agree that more or better practical training is needed. In contrast, livestock drivers remained undecided, indicating a possible gap in their understanding and knowledge in this area. This highlights the importance of providing comprehensive and targeted training programs for both slaughterhouse employees and livestock drivers to improve their ability to effectively comply with animal welfare standards. European Union legislation on the protection of animals at the time of killing requires that persons working with live animals have a certificate of competence [7]. In Slovenia, however, only employees of slaughterhouses have a certificate of competence, but not livestock drivers. This does not comply with the regulation, as the drivers unload live animals from the vehicle at the slaughterhouse. According to the World Organization for Animal Health [2], people involved in unloading, moving, lairage, caring, restraining, stunning, slaughtering, and bleeding animals play an important role in their welfare. Their competence must be acquired through formal training and practical experience in the slaughterhouse. After the theoretical part, employees in Slovenia must pass a written examination. However, the degree of concern for animal welfare depends on various parameters, such as gender, age, education, place of residence, dietary habits, etc. [13,14,15]. Improving the welfare of farm animals is costly. The simplest approach is to do nothing, but doing nothing can come at a cost. These costs come in the form of risks. Public concern about farm animal welfare has been studied over a long period of time [16,17,18,19], and there is evidence that public concern is increasing [16,17]. From different perspectives, some of the costs are one-time costs associated with changing infrastructure and practices, others are ongoing operational costs, and others are costs to which all companies in an industry must indirectly contribute. All of these costs are likely to be important factors in deciding which improvements should be made [20].

In addition, slaughterhouse employees showed a higher level of understanding of animal welfare compared to livestock drivers. They did not agree with the statement that farm animals have no feelings or only act out of fear. Slaughterhouse employees strongly affirmed that animals are sentient beings and showed a deeper understanding of their needs and behaviors. On the other hand, livestock drivers tended to agree with the statement that farmers should kill sick animals themselves to avoid expensive treatment or euthanasia costs. This discrepancy indicates a potential risk in the decision making of livestock drivers when they encounter animals that are not fit for transportation. The views of animal drivers could be influenced by cost considerations that could affect their judgment about the appropriate actions to take when dealing with animals that are unfit for transport. According to Grandin [21], “the most important thing is to have an animal fit for transport”. It is impossible to ensure the welfare of the animal during transportation if the animal is unfit for transport. However, the transportation of unfit animals is a common cause of non-compliance with animal health regulations [22]. The difference in knowledge and perspective between slaughterhouse employees and livestock drivers can be attributed to their different roles and responsibilities in the industry. Slaughterhouse employees who are directly involved in the day-to-day management and care of animals are likely to have more first-hand experience and knowledge of animal welfare. Livestock drivers, on the other hand, are primarily focused on transportation and may not have the same level of familiarity with animal welfare principles and practices. Encouraging collaboration and communication between slaughterhouse employees and livestock drivers can help create a shared commitment to animal welfare compliance throughout the supply chain. Where there are knowledge gaps in farm animal welfare, investment in research, development and knowledge enhancement is needed [20]. On the other hand, the welfare of farm animals can also have an impact on the quality of the end product. Extensive research has shown that meat quality improves when stress in cattle is reduced at slaughter [23,24,25,26]. Poor animal welfare during transportation and rearing can also lead to poorer product quality [27].

When comparing agreement with the statements on various topics relating to animal welfare, water and feed requirements, and health status, further differences emerged. The calculated *p*-values were statistically significant for three statements (*p* < 0.05). Firstly, the slaughterhouse employees were undecided on the statement that domesticated farm animals have different needs than their wild ancestors, while the livestock drivers agreed with this statement. This indicates a different understanding of the specific needs and characteristics of domesticated animals compared to their wild ancestors. Domestication is the process of adapting animals to live in close proximity to humans (domestic animals) or under human-made housing and husbandry conditions (livestock) [28]. Domestication can strongly influence the behavioral repertoire of an animal [29], i.e., selection pressure can change the behavior of animals during domestication [30]. There was also disagreement about the need for bedding in the slaughter pens, with slaughterhouse employees being undecided, indicating a lack of consensus among them, while livestock drivers tended to disagree. This suggests that they may have different views on the importance of bedding for animals in slaughter pens. If the animal is strongly motivated to perform behaviors that are no longer possible due to the lack of bedding, welfare problems may arise [31]. An example is the rooting behavior of pigs on concrete floors without bedding [32] while waiting overnight for slaughter. If the animal cannot perform desired behaviors because the required substrate is not available, this can lead to frustration [33]. Undecided responses in this case may indicate that further clarification or information on this issue is required. In the section on animal welfare, differences were found between opinions on mobile slaughterhouses and regular slaughterhouses. Slaughterhouse employees seemed more open-minded about both methods and were willing to consider alternative methods. One statement, for example, read: “Mobile slaughterhouses mean less suffering for the animals because the very stressful transportation for the animals is eliminated. Overall, the slaughterhouse employees agreed with this statement. However, the livestock drivers’ responses were a mixture of possible answers, ranging from strongly disagree to strongly agree. This can be attributed to the fact that slaughterhouse employees are familiar with the different approaches and the potential advantages and disadvantages of each method. The indecisiveness of the livestock drivers, on the other hand, indicates a lack of experience or knowledge of the differences between these approaches. There are no mobile slaughterhouses in Slovenia. The main reason for this is that farms are located close to at least one slaughterhouse, but not further than 100 km away [34]. The second statistically significant statement was whether animals are sentient beings. Slaughterhouse employees agreed more strongly with this than livestock drivers. The third statistically significant statement was disagreement about the impact of disease on animal welfare. Slaughterhouse employees disagreed with the statement that diseases do not affect animal welfare. This response indicates that they recognize the negative impact of disease on animal welfare and the importance of disease prevention and management. In addition, slaughterhouse employees are taught about the concept of five freedoms as part of their theoretical training for the certificate of competence, and the third concept is freedom from pain, injury or disease [35]. In contrast, livestock drivers were undecided, indicating a possible lack of knowledge as they are not trained according to Regulation (EC) 1099/2009. In Slovenia, farmers who achieved more than the required animal welfare standards in 2021 were supported by the Decree on Animal Welfare Measures under the Rural Development Program in the Republic of Slovenia for the period 2014–2020 [36]. As already mentioned [2], the transportation and housing of farm animals also play a very important role in animal welfare. For slaughterhouses, the one-time costs associated with changing infrastructure and converting practices can be a major burden on the business [20]. Therefore, the next step could be to extend animal welfare legislation to financially support better transportation practices and modern slaughterhouse equipment. The most concerning finding was that both slaughterhouse employees and livestock drivers were mostly undecided about the statement “‘biosecurity’ means that the feed for the animals is sanitary”. At a time when African Swine Fever (ASF) and Avian Influenza (AI) are threatening livestock populations, biosecurity is one of the most important preventative measures. It is very likely that trucks were a major factor in the spread of ASF in China [37], and trucks are also considered a potentially important risk for the spread of ASF in and around Europe [38]. Ssematimba [39] and other researchers investigated biosecurity measures in poultry farming and found that the main risk factors for the spread of AI are the movement of birds during thinning and restocking, most human movement when accessing the houses, and proximity to other poultry farms.

This study shows differences between the veterinarians’ assessments and the self-assessment of animal welfare by slaughterhouse employees and livestock drivers. In general, the veterinarians gave lower average scores than the participants, with the exception of one item (dust in the pan/lairage). Slovenian slaughterhouse employees give high priority to all parameters of animal welfare. This is consistent with the results of other studies conducted worldwide, which show that people consider animal welfare and animal welfare laws to be important, want better animal welfare, and consider animal welfare to be an important social issue [14,16,17]. The fact that the highest self-assessed scores were obtained for the parameters of housing conditions and environmental conditions in the stables shows that Slovenian slaughterhouse employees attach great importance to these aspects of animal welfare. Lairage refers to the area where the animals are kept before slaughter and it is crucial that the conditions in this area are conducive to animal welfare. The high level of self-assessment indicates that the slaughterhouse employees interviewed in the study recognize the importance of appropriate housing and environmental conditions at this stage of the animals’ journey. The high importance placed on lairage conditions and environmental conditions in the pens may be due to the recognition that the experiences animals have during this period can have a significant impact on their overall welfare and quality of life. Adequate ventilation, heating systems, and sufficient space are key factors that help to minimize stress, discomfort, and potential health problems for the animals [2,13,14,15]. The high self-assessment of these parameters also reflects the recognition of legal and ethical obligations related to animal welfare in the pre-slaughter phase. This indicates that slaughterhouse employees are aware of the importance of creating a favorable environment for animals in the pre-slaughter period.

The results in transportation also show a discrepancy between the experts’ observations and the participants’ self-assessments. In most cases, the experts’ observations resulted in lower scores than the participants’ self-assessments, indicating a possible difference in perception or awareness of animal welfare during transportation. However, for two parameters, the experts’ observations largely agreed with the participants’ self-assessments, particularly for environmental conditions, meaning that both experts and participants recognized the importance of adequate ventilation of the transport vehicle for animal welfare. However, a statistically significant difference (*p* < 0.05) was found for the parameter of the presence of thermometers and hydrometers. Livestock drivers indicated that temperature and humidity are important when transporting animals to the slaughterhouse. However, there are no procedures in place in case the temperature or humidity affects the welfare of the animals during transportation. The Dutch company Vion, for example, reduces the density of pigs on the vehicle when the outside temperature is 27 °C and prohibits the transportation of live animals when the temperature is 35 °C or higher [40]. One driver’s response that he did not care if he received a penalty for improperly transporting live animals to the slaughterhouse may have influenced his lower overall self-assessment score. This suggests that livestock transport drivers may not be aware that improper driving can cause stress and discomfort to animals, which can lead to a lower level of animal welfare. Transporting live animals is a critical stage in the overall animal welfare process and it is essential to ensure that animals are handled and transported in a way that minimizes stress, discomfort, and the risk of injury. Penalties for improper transportation or handling of animals should enforce regulations and promote proper handling of animals during transport [2,41].

## 5. Conclusions

The main findings of our study show that there is a difference between the perception of animal welfare by slaughterhouse employees and livestock drivers. Slaughterhouse employees have a stronger opinion and understanding than livestock drivers. This is probably due to a lack of knowledge about animal welfare, as livestock drivers do not have a certificate of competence according to Regulation (EC) 1099/2009. The employees of Slovenian slaughterhouses and the livestock drivers give high priority to animal welfare in the lairage and on transport. However, the assessment of the veterinary experts was almost always lower than that of the slaughterhouse employees and livestock drivers, which shows that there is room for improvement. Overall, these findings suggest that both slaughterhouse employees and livestock drivers in Slovenia need to be better trained and made aware of animal welfare practices and regulations.

Improved training programs should address the identified knowledge gaps and promote a deeper understanding of animal welfare issues across the industry. It is understandable that improving the welfare of farm animals in slaughterhouses through better employee training and new equipment is costly. But there can also be high costs associated with doing nothing, e.g., fines from official veterinarians and the possible loss of special certificates, e.g., Hilton Foods, McDonald’s, and market losses as a result. By improving the knowledge and perspective of all stakeholders, we can strive for better animal welfare outcomes in food production and transportation as well as better meat quality and food safety.

## Figures and Tables

**Figure 1 animals-14-00443-f001:**
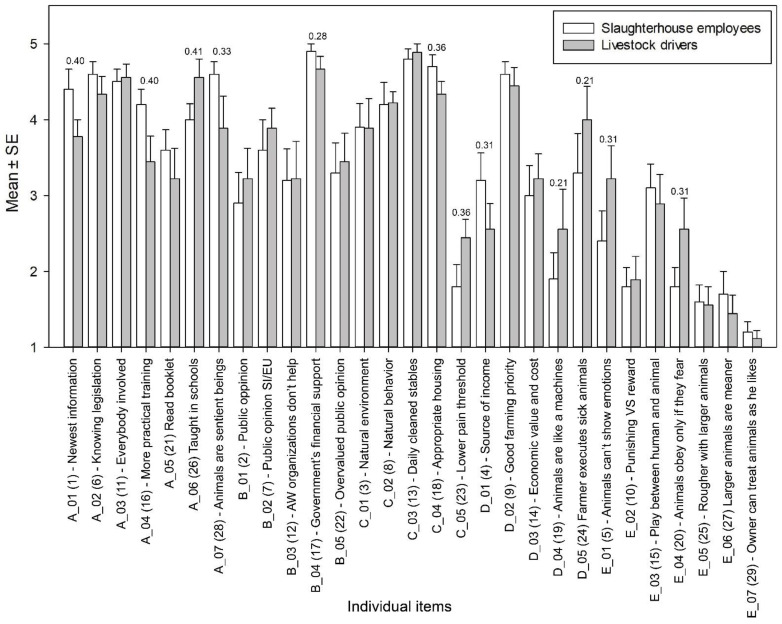
Comparison between slaughterhouse employees and livestock drivers in relation to animal welfare beliefs on the topics of knowledge of animal welfare laws (A), influence of public opinion on animal welfare (B), animal welfare (C), value of animals to humans (D) and relationship between humans and animals (E) (only effect size values > 0.20 are listed).

**Figure 2 animals-14-00443-f002:**
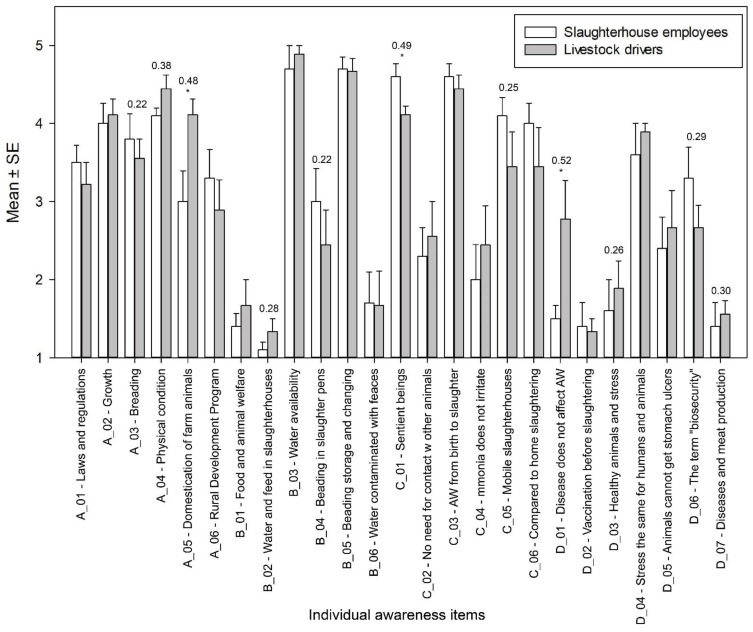
Comparison of animal welfare awareness between slaughterhouse employees and livestock drivers on the topics of animal welfare (A), water and feed requirements (B), the welfare of animals (C), and health status (D) (only effect size values > 0.20 are listed; * *p* < 0.05).

**Figure 3 animals-14-00443-f003:**
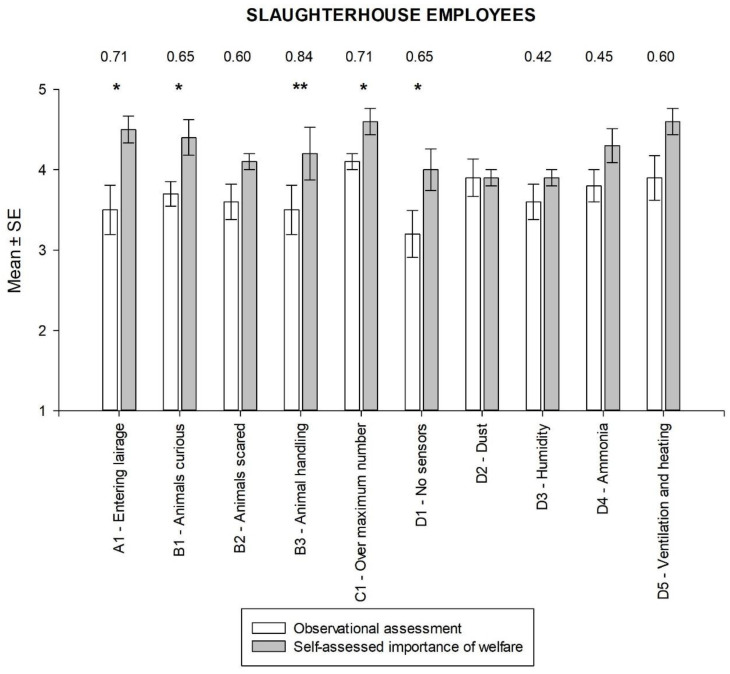
Comparison of perceived and actual animal welfare in the slaughterhouse on topics of General impression (A), Animal behavior (B), Lairage conditions (C), and Environmental conditions (D) (only effect size values > 0.20 are listed; * *p* < 0.05; ** *p* < 0.01).

**Figure 4 animals-14-00443-f004:**
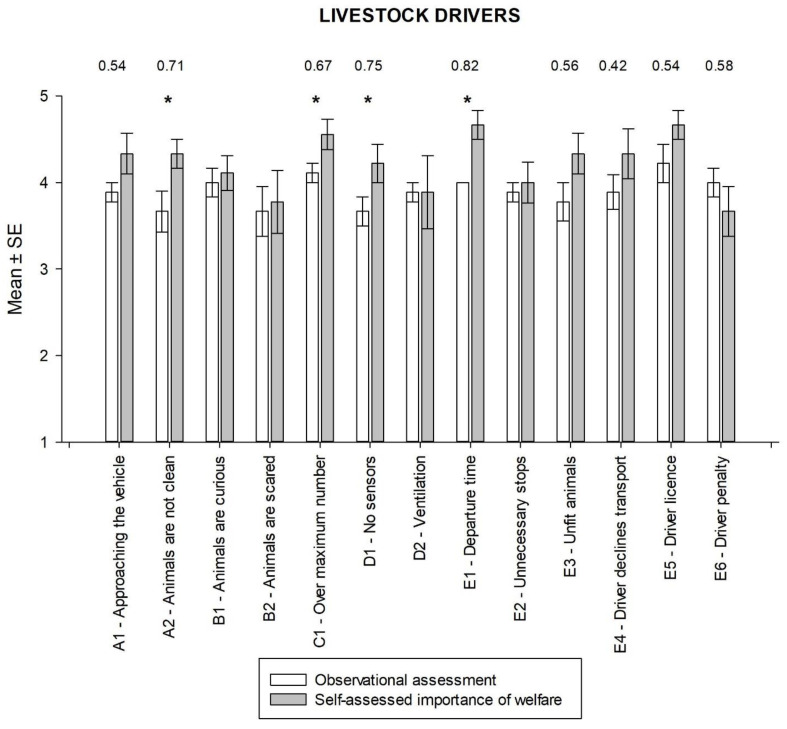
Comparison of perceived and actual animal welfare standards during transport on topics of General impression (A), Animal behavior (B), Transport conditions (C), Environmental conditions (D), Regulation compliance (E) (only effect size values > 0.20 are listed; * *p* < 0.05).

**Table 1 animals-14-00443-t001:** Visits of slaughterhouses (*n* = 10) in regional offices in Slovenia, categories of animals approved for slaughter, stunning methods and slaughter line speed.

Slaughterhouse Employees/Livestock Drivers	Region for Official Control	Animal Species Approved for Slaughter/Animals Transported on the Truck	Stunning Method of the Slaughterhouse	Slaughter Line Speed
Slaughterhouse employees	R1	Turkeys	Electrical waterbath	1100 turkeys per hour
	R5	Bovine, caprine, ovine, horses	Penetrative captive bolt device and head-only electrical stunning	7 bovine animals per hour 20 sheep or goats per hour
	R3	Bovine, porcine, horses	Penetrative captive bolt device and carbon dioxide at high concentration	25 bovine animals per hour 100 pigs per hour
	R10	Chicken broilers	Carbon dioxide in two phases	8600 broilers per hour
	R1	Bovine, porcine	Penetrative captive bolt device and head-only electrical stunning	25 bovine animals per hour 30 saws per hour
	R2	Bovine, porcine	Penetrative captive bolt device and carbon dioxide at high concentration	30 bovine animals per hour 100 pigs per hour
	R3	Porcine	Carbon dioxide at high concentration	110 pigs per hour or 80 piglets
	R5	Chicken broilers	Electrical waterbath	4000 broilers per hour
	R9	Chicken broilers	Electrical waterbath	4000 broilers per hour
	R9	Porcine	Carbon dioxide at high concentration	100 pigs per hour
Livestock drivers	R1	Turkeys	NA	NA
	R5	Bovine (young bulls)	NA	NA
	R3	Bovine (milk cows)	NA	NA
	R10	Broilers	NA	NA
	R1	Bovine (young bulls)	NA	NA
	R3	Porcine	NA	NA
	R5	Chicken broilers	NA	NA
	R9	Chicken broilers	NA	NA
	R9	Porcine	NA	NA

NA = not applicable.

**Table 2 animals-14-00443-t002:** Gender, age, and level of education of the participants.

Category	Slaughterhouse Employees	Livestock Drivers
**Gender**		
Male	7	9
Female	3	-
**Age**		
<40 years	2	2
>40 years	8	7
**Education**		
Primary school	1	1
High school	6	8
Professional degree	2	-
Unified Master’s program	1	-

## Data Availability

The data presented in this study are available on request from the corresponding author. The data are not publicly available due to [protection of personal data].

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
