# Peer review of "Farm Animal Welfare during Transport and at the Slaughterhouse: Perceptions of Slaughterhouse Employees, Livestock Drivers, and Veterinarians"

_animals, 2024, doi:10.3390/ani14030443_

Round 1
Reviewer 1 Report
Comments and Suggestions for Authors
Very good job. Just some minor things that need clarification.
78-79 Needs rewording, does not make sense.
81 Add "truck" drivers
88 Replace "that" with "who"
91 Is this per day or week?
Table 1 Not sure if this is necessary. R2 the same place for employees as livestock drivers? Why different species?
157 "their"
219 Drop the second "mostly"
Fig 2 Title - What are the four different topics?
267 He "already" received a penalty? Not clear
333 Drop "Grandin" or include everyone else
349 Need for bedding may be related to who cleans the pens
Comments on the Quality of English Language
Just a few minor corrections, otherwise very good.
Reviewer 2 Report
Comments and Suggestions for Authors
The present work highlighted the importance of awareness and training among the personnel involved in the meat production industry. The study involved a survey based on 10 slaughterhouse employees and 9 livestock transport drivers. The hypothesis is strong and well-established. The language is easy to understand.
I have the following observations-
i. L34- please mention the place/ region and time of the study, as this knowledge may vary from place to place
ii. L20: the expert------livestock driver, please rewrite with more clarity
iii. L30: Sentence ‘Several parameters of animal welfare, such as health status, animal behavior, lairage or transport vehicle conditions, and driver regulation compliance’ incomplete
iv. So, for self-assessment- no scale used
v. L34: please mention the number of participants used in the study in the 9 slaughterhouses and the duration of the survey.
vi. L79: Authors may mention the application of the study for improving the hypothesis.
vii. Methodology: Need to add further details in the selection and design of sampling, such as - how many participants are in the slaughterhouses. The questionnaire was posted/ handed over to the participants, and they filled in their own or authors recorded, questions in their local language along with English translation, also slaughter practices in the selected slaughterhouse such as by stunning (type), modern slaughterhouse, etc.
viii. For 10 slaughterhouses and 1 employee from each so 10 slaughterhouse employees and 9 livestock drivers? Please add more clarity on these matters.
ix. Statistical analysis: Appropriate
x. L450: means at the high cost of animal welfare? Please explain.
Comments on the Quality of English LanguageThe language is fine and need only minor editing.
Reviewer 3 Report
Comments and Suggestions for Authors
The study conducted by Lipovšek et al. aimed to assess the perceptions of slaughterhouse employees and drivers concerning animal welfare during transportation. The research employed a well-structured questionnaire and presented its findings in a simple and easily comprehensible manner. However, a significant limitation of the study is the sample size. For observational studies, a minimum sample size of at least 30 subjects is generally recommended to ensure a meaningful assessment. With only 19 participants in the current study, the small sample size represents a major limitation, raising concerns about the validity of the research findings. The authors are strongly encouraged to increase the number of participants in order to enhance the validity and reliability of the study's findings. The authors are strongly encouraged to increase the participant size to improve the validity and reliability of the study's findings. As it stands, in its current form, the study is not recommended for further review.
Comments on the Quality of English LanguageThere are some grammatical errors with missing predicates, for instance, in the sentence found in lines 30-31.
Reviewer 4 Report
Comments and Suggestions for Authors
Accept - Minor Revision - This paper contains important information on how people perceive animal welfare. To make it acceptable for publication will require a much more complete explanation of how the slaughter houses and the people were selected.
Line 14 - Insert beef, pork and poultry before the word slaughterhouse. It is important for readers to know that three different species were studied.
Line 29 - Insert beef, pork and poultry before the word slaughterhouse.
Line 82 - Please describe the training materials that are used for studying for the certificate of competence. In most countries, training materials include both regulations and information on animal behavior stunning, and animal handling. Please provide references for training materials used in Slovenia.
Line 87 - What was the method for selecting one livestock driver for each slaughterhouse? Was the drive randomly chosen or was he the first driver who drove up when the researchers were present? Please explain.
Line 96 - If there were three employees that were available for being surveyed how did you choose the one employee that your surveyed. In all survey work, it is extremely important to fully describe sampling methods.
Table 1 - Add a column which shows the approximate hourly slaughter line speed for each slaughterhouse. This information provides important information for making comparisons with other studies.
Line 105 - It is really good that you are going to publish the entire survey.
Line 122-125 - How were the veterinarians chosen to make observations in each slaughterhouse? Please describe the training they had. If you used published training materials, the references should be added.
Line 204 - Translation error? What does "breading" mean? Do you mean bedding?
Line 220 - Change irreproachable to sanitary.
Line 298 - If the certificate of competence is obtained through practical experience, do the employees take a test? In many countries, employees have to take an approved training program and pass a test.
Line 346 - Do you have any discussions about different responses for different species?
Line 370 - Were there any species differences between poultry (white meat) and cattle/pigs (red meat) on the sentience question?
Line 449 - This reviewer completely agrees with the statement about the need for more training. Both employees and drivers need training on both the regulations and the behavior of the animals.
Comments on the Quality of English Language
No comment
Round 2
Reviewer 3 Report
Comments and Suggestions for Authors
The study conducted by Lipovšek et al. aimed to assess the perceptions of slaughterhouse employees and drivers concerning animal welfare in Slovenia. The research employed a well-structured questionnaire and presented its findings in a simple and easily comprehensible manner. While the study's applicability is constrained by its scope, it holds potential importance for policymakers in the country and under similar circumstances. A notable limitation is the study's sample size, limiting its applicability to a broader audience despite its valuable information. To address the sample size issue, consider portraying the study as a case study of larger slaughterhouse in Slovenia. Additionally, it is suggested to include this aspect in the title, for example: 'Case Study of Farm Animal Welfare During Transport and at the large Slaughterhouses in Slovenia. Additional feedback is provided below to further improve the manuscript's quality.
Line 21-2: What results were lower?
Line 32–33: This sentence should be moved after the next sentence where the authors described that the study consisted of two parts.
Line 35: Please consider removing the sentence 'The results were compared with each other' from the abstract."Top of Form
Line 49: Please provide a reference for the description of “Animal Welfare”.
Materials and methods
For improved organization and flow, the first paragraph should detail the study area, including the total population of slaughterhouses and the number of employees. The second paragraph should cover the inclusion criteria for the survey and the sampling strategy. Additionally, please include information on the study's limitations concerning the sample size.
Lines 109-122: The sampling strategy is not clearly defined. Please provide a more detailed explanation of the sampling methodology in this section.
Line 111: From my understanding, Slovenia has a total of 83 slaughterhouses, of which only 25 are classified as large slaughterhouses. It appears that the study was conducted only on 10 large slaughterhouses. Please confirm and clarify this point for better comprehension.
Results
The comparison of scores between slaughterhouse employees and the experts was entirely absent from the results section. It is crucial to include this comparison in the results section, as it was a major point emphasized in the abstract.
Conclusion
The conclusion needs to be updated to provide a concise summary of the findings rather than primarily serving as recommendations. Please revise to focus on summarizing the study's key outcomes. It is encouraged to move the recommendations after the conclusion paragraph for better organization and flow.
Author Response
We have made some of the changes.